# Association of In-School and Electronic Bullying with Suicidality and Feelings of Hopelessness among Adolescents in the United States

**DOI:** 10.3390/children10040755

**Published:** 2023-04-21

**Authors:** Tran H. Nguyen, Gulzar Shah, Maham Muzamil, Osaremhen Ikhile, Elizabeth Ayangunna, Ravneet Kaur

**Affiliations:** 1Department of Interdisciplinary Health Sciences, College of Allied Health, Augusta University, Augusta, GA 30912, USA; 2Department of Health Policy and Community Health, Jiann-Ping Hsu College of Public Health, Georgia Southern University, Statesboro, GA 30460, USA; 3Department of Education, Kinnaird College for Women’s University, Lahore 54000, Pakistan; 4Department of Family and Community Medicine, College of Medicine, University of Illinois, Chicago, IL 60612, USA

**Keywords:** bullying, in-school bully, cyberbullying, suicide-related behaviors, suicidality, depressive symptoms, hopelessness

## Abstract

Background: Suicide-related behaviors increasingly contribute to behavioral health crises in the United States (U.S.) and worldwide. The problem was worsened during the COVID-19 pandemic, especially for youth and young adults. Existing research suggests suicide-related behaviors are a consequence of bullying, while hopelessness is a more distal consequence. This study examines the association of in-school and electronic bullying with suicide-related behavior and feelings of despair among adolescents, adjusted for sociodemographic characteristics, abuse experience, risk-taking behaviors, and physical appearance/lifestyles. Method: Using Chi-square, logistic regression, and multinomial logistic regression, we analyzed the US 2019 Youth Risk Behavior Surveillance System (YRBSS) national component. The YRBSS includes federal, state, territorial, and freely associated state, tribal government, and local school-based surveys of representative sample middle and high school students in the US. The 2019 YRBSS participants comprised 13,605 students aged 12 to 18 years and roughly equal proportions of males and females (50.63% and 49.37%, respectively). Results: We observed a significant association (*p* < 0.05) between being bullied and depressive symptoms, and the association was more vital for youth bullied at school and electronically. Being bullied either at school or electronically was associated with suicidality, with a stronger association for youth who experienced being bullied in both settings. Conclusion: Our findings shed light on assessing early signs of depression to prevent the formation of suicidality among bullied youth.

## 1. Introduction

Bullying, repeated violence stemming from power imbalances, is perpetrated through various means, including social exclusion, verbal or/and physical abuse, rumor-mongering, and peer coercion [1]. The chronicity of these bullying activities and longer durations of exposure have been shown in longitudinal studies to increase the likelihood of suicidal ideation and suicide attempt [2,3]. There is also evidence that other coexisting risk factors, such as depression, low self-esteem, anxiety, loneliness, and hopelessness, may mediate the relationship between bullying and suicidal tendencies [4,5,6]. Hopelessness is a manifestation of depression, including helplessness as an essential element of hopelessness [5]. Individuals develop negative thought patterns with self-blaming, viewing the cause of events as unchangeable, and overgeneralizing weaknesses in several areas. Youths experiencing bullying may assign attributes and importance to each bullying episode, including the reason for its occurrence, what it represents to the self and its potential consequences. Repetition of these processes from constant bullying leads to the development of hopelessness, subsequently increasing the risk for the development of suicide-related behavior, including suicidal ideation and suicide attempt [5].

Suicide-related behaviors contribute significantly to behavioral health crises in the United States and worldwide, particularly during the COVID-19 pandemic and especially for youths and young adults [7,8,9]. Evidence points to suicide-related behaviors as consequences of bullying, while hopelessness is a more distal consequence of bullying [10,11,12,13,14]. Youths with elevated suicidal ideation are more likely to exhibit hopelessness than those with subclinical suicidal ideation [15,16]. Researchers worldwide have studied pathways between bullying, feelings of hopelessness, and suicide-related behavior. A meta-analysis by Holt and colleagues indicated that bullying victimization, bullying perpetration, and bully/victim status were risk factors for suicide-related behaviors [17]. Similarly, a European study showed that low peer support increased the associations between verbally bullied victimization and suicide ideation [18]. A study in Sweden showed self-harm and suicidal ideation were negative consequences of a high degree of long-term bullied victimization [19]. According to the longitudinal study by Karlsson and colleagues, parental affection and teacher support were two essential protective factors against bullying. In contrast, the risk factors included anxiety, depression, and somatic symptoms [20]. Additional studies show that the perception of social support and guidance from experts and family has been shown to protect against suicide attempts, while hopelessness and bullying were found to be risk factors for suicide-related physical violence [21,22,23].

Bullying has typically occurred at school during school hours. However, in the era of the internet, electronic bullying or cyberbullying is commonly perpetrated through electronic communication devices, such as cell phones, email, instant messaging, and social media sites, and at all hours of the day/night [1,24]. Compared to in-school bullying, cyberbullying has more potent effects in predicting suicide-related behavior [4,17,25]. The impact of cyberbullying is often complicated by the perpetrator’s anonymity and the potential frequency (e.g., the potential to bully 24 h a day vs. in select settings). When assessing the relationship between bullying and suicide-related behavior in youth presenting to the emergency department (ED), Alavi et al. found that 77% of the adolescents had experienced bullying, and 68.9% had suicide ideation at the ED visits [4]. The study was controlled for age, gender, grade, psychiatric diagnosis, and abuse, and found that the history of bullying was the strongest predictor of suicide ideation. Moreover, cyberbullied youths were 11.5 times more likely to have suicidal ideation documented at ED visits than physically bullied ones.

Much emphasis has been given to bullying and mental health across the literature, while less attention focuses on the association between bullying and suicide-related behavior. There is also a paucity of studies relating bullying with feelings of hopelessness. This study aims to contribute to understanding the association of in-school and electronic bullying with suicide-related behavior and feelings of despair among adolescents. A comprehensive approach that simultaneously targets multiple risks and protective factors is critical to having a broad and continued impact on youth violence. We examined the association of in-school and electronic bullying with suicide-related behavior and feelings of despair among adolescents, adjusted for sociodemographic characteristics, abuse experience, risk-taking behaviors, and physical appearance/lifestyles.

## 2. Materials and Methods

### 2.1. Data Source

We analyzed the 2019 Youth Risk Behavior Surveillance System (YRBSS) national component. The YRBSS includes federal, state, territorial, and freely associated state, tribal government, and local school-based surveys of representative sample middle and high school students in the United States (U.S.). These surveys are conducted biennially to monitor priority health-risk behaviors that contribute markedly to the leading causes of death, disability, and social problems, as well as the prevalence of obesity and asthma among youth in the U.S. The Centers for Disease Control and Prevention (CDC) conducts the National YRBS, using a three-stage cluster design to represent 9th- to 12th-grade students in public and private schools. The 2019 YRBS included 13,677 student responses and achieved an overall response rate of 60.3%. The YRBSS website at www.cdc.gov/yrbss (accessed on 17 March 2022). contains additional methodology and data analysis information.

### 2.2. Measures

We were interested in two dependent variables to examine the association between bullied youth and their mental health, reflecting (1) depressive symptoms and (2) suicidality. The dependent variable measuring the youth’s depressive symptoms were operationalized through the survey item asking, “During the past 12 months, did you ever feel so sad or hopeless almost every day for two weeks or more in a row that you stopped doing some usual activities?” The response choice was dichotomous as (no) or (yes). The second dependent variable, “suicidality”, covers suicidal ideation (painful thoughts about taking one’s own life), suicide plans, and suicide attempts. We combined three survey items questioning whether they ever considered, planned, and attempted suicide into a categorized variable. The survey items in this regard were “During the past 12 months, (1) did you ever seriously consider attempting suicide? (2) did you make a plan about how you would attempt suicide? and (3) how many times did you actually attempt suicide?” If the youth reported more than one category of suicidality, we recorded their responses to the highest level, ranging from not suicidal, considering suicide, planning, to attempting suicide.

We designated three independent variables. The first two independent variables of interest were derived from the survey asking the youth (1), “Have you ever been bullied on school property?” And (2) “Have you ever been electronically bullied? (Counting bullying through e-mail, chat rooms, instant messaging, websites, or texting)”. Their response choices were either (no) or (yes). We combined these two survey items into a 3-categorized variable, reflecting the number of bullying types the youth suffered. The three categories were (i) neither bullied at school nor electronically, (ii) either, or (iii) both. Sociodemographic characteristics such as age, gender, and race/ethnicity were used as control variables. In addition, we include covariates reported in the literature to be associated with youth suicidality. The factors were the youth’s experience of abuse (as in sexual abuse and physical dating violence), risk-taken behaviors (used marijuana/alcohol and had an early sexual relationship), physical appearance (obesity), and physical lifestyles (spent a long time on digital games, was physically inactive, and got enough sleep) [26,27,28,29,30,31,32,33,34]. Table 1 provides detailed descriptions of the covariates.

### 2.3. Data Analysis

The statistical software used in all analyses was the STATA version 15.0 (StataCorp LLC, College Station, TX, USA). We performed descriptive statistics of variables as appropriate. We used Chi-square analyses to compare the youth’s mental state between the bullied and not bullied. We conducted a logistic regression to examine the association between youth being bullied at school or electronically and the depressive symptom of sadness or hopelessness. We ran a multinomial logistic regression to understand the relationship between the number of bully types and youth suicide ideation. Both logistic regressions were adjusted for sociodemographic characteristics, abuse experience, risk-taking behaviors, and physical appearance/lifestyles. Statistical tests were at the significance level of α = 0.05. We utilized the sampling weights developed by the YRBSS in all the analyses to reflect the multistage sampling design of the survey.

## 3. Results

Table 2 presents the description of the study variables. A total of 13,605 middle and high school students aged 12 to 18 years responded to the 2019 YRBSS. The respondents’ mean age was 15 years (standard deviation = 1.7). The proportion of females (49.4%) and males (50.6%) was almost equal. The racial makeup was 51.2% White, 12.2% Black, 9.2% Hispanic, 5.1% Asian, 0.6% American Indian or Alaska Native, and 0.3% Native Hawaii or other Pacific Islander. Also included in the sample were 16.9% multiracial Hispanic and 4.5% multiracial non-Hispanic.

Nearly 19.5 % of respondents reported being bullied at school, while 15.7% electronically. Most respondents (75.3%) were not bullied, 14.6% were bullied at school or electronically, and 10.1% were both. Thirty-seven percent (36.7%) of respondents reported feeling sad or hopeless, whereas 18.5% considered suicide, 15.7% planned, and 8.9% attempted suicide at least once.

A small portion of youth had experienced sexual abuse (7.3%) and physical dating violence (8.2%). As for a risk-taking lifestyle, while 21.8% of youth reported using marijuana and 29.22% used alcohol, only 3.0% reported having a sexual relationship before the age of 13. Approximately 15.5% of respondents reported being obese. Whereas 22.1% of respondents reported having more than 8 h of sleep a day, almost equal proportions reported being physically active (44.1%) and spending more than 3 h on digital games and computers (46.1%).

Table 3 shows the Chi-square analysis of the correlation between being bullied with depressive symptoms and suicidality. The bullied youth had significantly higher depressive symptoms and suicidality values than their counterpart either at school or electronically. Sixty-two percent (62.1%) of those bullied at school felt sad or hopeless, 39.4% had suicidal thoughts, 32.4% planned suicide, and 21.3 % attempted suicide at least once. Youth who reported being bullied electronically exhibited slightly worse suicidality than those bullied at school, with 65.4% feeling sad or hopeless, 42.1% having suicidal thoughts, 34.7% planning suicide, and 23.4% attempting suicide at least once.

Table 4 displays the logistic regression analysis of the association between being bullied and depressive symptoms after controlling for various variables. In contrast to their counterpart, youth who reported being bullied at school or electronically exhibited higher odds, 2.43 times (CI, 2.05–2.87), of depressive symptoms. The odds increased for youth who suffered bullying both at school and electronically; they were 3.46 times (CI, 2.84–4.22) more likely to exhibit depressive symptoms than the non-bullied youth.

Covariates reflecting if a youth experienced abuse, used alcohol and marijuana, spent a long time on digital devices, and appeared obese, were positively associated with the youth’s depressive symptoms. Youth who experienced sexual abuse and physical dating violence were 2.83 (CI, 2.22–3.60) and 3.12 (CI, 2.40–4.05) times, respectively, more likely than those who did not. Youth who exhibited risk-taken behaviors using alcohol and marijuana were more likely to show depressive symptoms than those who did not respond by 1.18 (1.03–1.35) and 1.52 (CI, 1.31–1.75) times. The same trend was observed with youth who spent more time on digital devices and appeared obese, 1.68 (CI, 1.49–1.89) and 1.24 (CI, 1.09–1.41) times more.

Gender, ≥8 h of sleep per day, and being physically active at least 60 min per day on ≥5 days were negatively associated with youth’s depressive symptoms. Males were 0.51 (CI, 0.45–0.58) times as likely to have depressive symptoms as females. Youth who got ≥8 h of sleep per day and were physically active at least 60 min per day on ≥5 days were 0.46 (CI, 0.39–0.55) and 0.72 (CI, 0.64–0.80) times, respectively, as likely as their counterparts to have depressive symptoms. Age, race/ethnicity, and sex the first time before 13-year-old were not statistically significantly associated with youth depressive symptoms.

Table 5 shows the multinomial regression of the association between the number of bullying types and youth suicidality after adjusting for the designated control variables. The displayed relative risk ratios indicated that those who suffered either or both bullying types were significantly at greater risk of considering suicide (RRR_either_ = 1.59 (CI, 1.16–2.18); RRR_both_ = 2.70 (CI, 1.96–3.73)), planning (RRR_either_ = 2.09 (CI, 1.64–2.65); RRR_both_ = 3.09 (CI, 2.40–3.98)), and attempting (RRR_either_ = 2.10 (CI, 1.60–2.74); RRR_both_ = 3.82 (CI, 2.92–4.99)) and at lower risk of not having suicidality than those who were not bullied.

For other covariates, gender, experience of sexual abuse, experience of physical dating violence, use of marijuana, video/computer games or computers more than 3 h a day, and appearance obese were significant positive predictors. The relative risk ratios indicated that those who reported (yes) to those items were at greater risk than their counterparts to considering suicide (RRR_sex-abused_ = 2.04 (CI, 1.41–2.94); RRR_date-violence_ = 1.57 (CI, 1.04–2.37); RRR_marijuana_ = 1.77 (CI, 1.36–2.32); RRR_digital_ = 1.72 (CI, 1.36–2.18); RRR_obese_ = 1.80 (CI, 1.42–2.29)), planning suicide (RRR_sex-abused_ = 2.50 (CI, 1.87–3.35); RRR_date-violence_ = 2.01 (CI, 1.47–2.74); RRR_marijuana_ = 1.78 (CI, 1.44–2.21); RRR_digital_ = 1.69 (CI, 1.40–2.03); RRR_obese_ = 1.48 (CI, 1.21–1.79)), and attempting suicide (RRR_sex-abused_ = 4.36 (CI, 3.29–5.78); RRR_date-violence_ = 2.26 (CI, 1.65–3.10); RRR_marijuana_ = 2.29 (CI, 1.81–2.88); RRR_digital_ = 1.50 (CI, 1.22–1.84); RRR_obese_ = 1.91 (CI, 1.55–2.35)), relative to having no suicidality. In contrast, gender, getting at least 8 h of sleep per day, and being physically active for at least 60 min per day for a minimum of 5 days were significant negative predictors. The relative risk ratios indicated that those who reported (yes) to those items were at lesser risk of considering suicide (RRR_gender_ = 0.61 (CI, 0.50–0.75); RRR_well-sleep_ = 0.43 (CI, 0.32–0.58); RRR_physical-active_ = 0.55 (CI, 0.42–0.71)), planning suicide (RRR_gender_ = 0.47 (CI, 0.40–0.55); RRR_well-sleep_ = 0.39 (CI, 0.26–0.58)), and attempting suicide (RRR_gender_ = 0.64 (CI, 0.51–0.815); RRR_well-sleep_ = 0.40 (CI, 0.28–0.56); RRR_physical-active_ = 0.63 (CI, 0.51–0.79)), relative to having no suicidality, than their counterparts. Age, race/ethnicity, and having sex first time before 13-year-old showed no significant association with youth suicidality.

## 4. Discussion

Youth suicide has become a growing public health concern in the past decade, as youth suicidality linked in some way to bullying has become common [35,36,37]. Federal public health agencies, violence prevention partners, and researchers have invested in learning more about the complex web of factors causing or contributing to these public health problems to design interventions for the prevention of bullying. Our study showed that during 2019, 1 in 18 youth reported considering suicide, 1 in 10 considered and planned suicide, and 1 in 13 considered, planned, and attempted suicide. Among youth respondents, while 1 in 7 reported being bullied either at school or electronically, 1 in 10 said they were bullied both at school and electronically. We also found that a third of the youth reported depressive symptoms, such as feeling sad and hopeless.

Bullying, depression, and suicidality are intertwined [37]. The association between being bullied and suicidality is well documented. Studies have also shown that individuals who exhibit depressive symptoms were more likely to experience suicidality [38]. Our study examined the association of being bullied with depressive symptoms and suicidality. The current study found that being bullied either at school or electronically was associated with suicidality, with a stronger association for youth who experienced being bullied in both settings. Moreover, the risks for suicidality among bullied youth significantly increased from an idea to a plan, and finally, an attempt. Understanding these associations is crucial in suicide prevention.

Our results show that age, race, and early sexual relationship were not associated with depressive symptoms. We also observed no significant associations of race and early sexual relationship with suicidality. Age shows a strong association with suicidality only in youth who planned and/or attempted suicide in certain age groups. In addition, alcohol use did not show an association with suicidality. Other than the relationships mentioned above, our findings are consistent with past studies among other factors such as the experience of abuse (as in sexual abuse and physical dating violence), risk-taking behaviors (used marijuana), physical appearance (obesity), and physical lifestyles (spent a long time on digital games, was physically active, and got enough sleep) [26,27,28,29,30,31,32,33,34].

Our results corroborate and extend past studies, showing significant associations of bullying with depression and suicidality [37,38,39,40,41]. In addition, this current study demonstrated that victimization via cyberbullying and in-school bullying was associated with depressive symptoms, which, in turn, gradually increased the risk for suicidal ideation, suicide planning, and suicide attempt. Our findings suggested that a goal of suicide prevention efforts concerning bullying should be systematic screening for early signs of mental distress, such as feelings of sadness and hopelessness. The correlation between being bullied and developing depressive symptoms such as sadness and hopelessness suggests that screening efforts should include sadness and hopelessness as red flags. These screening red flags for bullying victims may assist with timely intervention and prevention of negative feelings from turning into adverse ideas, plans, and actions. Our finding on this relationship implied that depressive symptoms should be recognized and evaluated for bullying before they develop into suicidality. Future studies may dwell on the intertwined relationship of bullying, depressive symptoms, and suicidality among youth. Perhaps depressive symptoms act as a mediation between bullying and suicidality. In addition, scholars need to expand research on bullying in the cyber setting, as cyberbullying has become an essential determinant in depressive symptoms and suicidality.

As we reported noteworthy results, our study had some limitations to consider when generalizing the results. We used secondary data that limited our ability to examine the relationship between depressive symptoms and suicidality with other relevant factors such as socioeconomic status, childhood abuse/neglect, time spent on social networks, the extent of the engagement activity in social networks, and social support from peers, teachers, and parents, to name a few. In addition, the nature of cross-sectional data constrained our ability to conclude the causation between being bullied and depressive symptoms and suicidality. Therefore, our results only support inferring associations. The use of 2019 YRBS subjects in our study may have included recall bias, as the data were based on self-reporting. However, YRBS investigators screened the information for the conflict response in logical terms to minimize recall bias.

Despite the limitations, using nationally representative data was our study’s strength in generalization. Nevertheless, our findings must be used cautiously amid the COVID-19 pandemic, which has put tremendous stress on the youth due to unprecedented infection control measures, such as nationwide school closures. The literature has documented the severe impact of the pandemic on youth mental health and well-being [41,42]. The following YRBSS cycle, which was conducted in the fall of 2021, represents the first data collected since the start of the COVID-19 pandemic. Studies that compare pre- and post-pandemic data are warranted to establish the short- and long-term impact that the COVID-19 pandemic imposed on American youth.

## 5. Conclusions

Our findings shed light on the relationship between being bullied and depression and suicidality among youth. Young victims of bullying experience ongoing and perpetual physical and mental health harm. To prevent youth from such harm, all stakeholders must make preventing bullying their top priority, including public health practitioners, schools, parents, social media administrators, and the cyber community. By employing a comprehensive approach that simultaneously targets multiple risks and protective factors, our study found that (1) being bullied either at school or electronically is associated with depressive symptoms and suicidality and (2) the risk of severe outcomes is greater for bullying victims experiencing multiple types of bullying. It is imperative that school officials/nurses/teachers, parents, and healthcare providers assess for early signs of depression to prevent suicidality in victims of bullying. More research is needed to develop effective interventions for reducing both in-school and cyberbullying.

## Figures and Tables

**Table 1 children-10-00755-t001:** Detailed description of covariates based on the 2019 Youth Risk Behavior Surveillance System (YRBSS) national component.

Covariate	Survey Item	Response Choice
Experience of sexual abuse	Have you ever been physically forced to have sexual intercourse when you did not want to?	(no) or (yes)
Experience of physical dating violence	Experience physical dating violence	(no) or (yes)
Risk-taken behaviors using marijuana/alcohol	Currently used marijuana	(no) or (yes)
Risk-taken behaviors having an early sexual relationship	Had sexual intercourse for the first time before the age of 13 years	(no) or (yes)
Physical appearance of obesity	Had obesity (students who were ≥95th percentile for body mass index, based on sex- and age-specific reference data from the 2000 CDC growth charts)	(no) or (yes)
Physical lifestyles of spending a long time on digital games	Played video or computer games or used a computer 3 or more hours per day (counting time spent on things such as playing games, watching videos, texting, or using social media on your smartphone, computer, Xbox, PlayStation, iPad, or other tablet, for something that was not schoolwork, on an average school day)	(no) or (yes)
Physical lifestyles of being physically inactive	Were physically active at least 60 min per day on 5 or more days (in any kind of physical activity that increased their heart rate and made them breathe hard some of the time during the 7 days before the survey)	(no) or (yes)
physical lifestyles of getting enough sleep	Got 8 or more hours of sleep (on an average school night)	(no) or (yes)

**Table 2 children-10-00755-t002:** Description of dependent, independent, and control variables (N = 13,677).

Variables		N ^a^ (Not Weighted)	% (Weighted)
**Dependent variables**			
Depressive symptom of feeling sad or hopeless (n = 13,490)	No	8538	63.3
	Yes	4952	36.7
Suicidal tendency severity (n = 13,537)	No suicidal tendency	7912	58.5
	Considering suicide	755	5.6
	Planning suicide	1306	9.6
	Attempting suicide	1018	7.5
**Independent variables**			
Bullied at school (n = 13,506)	No	10,870	80.5
	Yes	2636	19.5
Bullied electronically (n = 13,524)	No	11,400	84.3
	Yes	2125	15.7
Number of bully types—at school and/or electronically (n = 13,677)	Neither	10,298	75.3
	Either	1997	14.6
	Both	1382	10.1
**Control variables**			
Age (n = 13,600)	≤14 years old	1670	12.3
	15 years old	3369	24.8
	16 years old	3480	25.6
	17 years old	3218	23.7
	≥18 years old	1864	13.7
Gender (n = 135,520)	Female	6690	49.4
	Male	6862	50.6
Race (n = 13,253)	AIAN ^b^	86	0.6
	Asian	672	5.1
	Black	1616	12.2
	NH/PI ^c^	44	0.3
	White	6784	51.2
	Hispanic	1216	9.2
	Multiple Hispanic	2245	16.9
	Multiple non-Hispanic	591	4.5
Experienced sexual abuse (n = 12,141)	No	11,252	92.7
	Yes	889	7.3
Experienced physical dating violence (n = 8798)	No	8074	91.8
	Yes	724	8.2
Currently used marijuana (n = 13,352)	No	10,448	78.2
	Yes	2904	21.8
Currently used alcohol (n = 12,689)	No	8981	70.8
	Yes	3708	29.22
Sex first time before the age of 13 years (n = 12,075)	No	11,715	97.0
	Yes	361	3.0
Got ≥ 8 h of sleep per day (n = 13,157)	No	10,249	77.9
	Yes	2908	22.1
Physically active at least 60 min per day on ≥5 days (n = 13,287)	No	7427	55.9
	Yes	5860	44.1
Used video/computer games or used computer ≥3 h per day (n = 13,244)	No	7136	53.9
	Yes	6108	46.1
Obesity (n = 12,094)	No	10,223	84.5
	Yes	1871	15.5

^a^ Numbers that do not add up to the total (N) indicate missing values. ^b^ AIAN = American Indian or Alaska Native; ^c^ NH/PI = Native Hawaii or other Pacific Islander.

**Table 3 children-10-00755-t003:** Chi-square (χ^2^) analysis of the association of being bullied with mental states.

		Bullied at School			Bullied Electronically		
		No n (%)	Yes n (%)	χ^2^	No n (%)	Yes n (%)	χ^2^
**Feeling sad or hopeless**	No	7424 (69.6)	1009 (37.9)	<0.001	7738 (68.7)	727 (34.6)	<0.001
	Yes	3239 (30.4)	1655 (62.1)		3528 (31.3)	1372 65.4)	
**Considering suicide**	No	9132 (85.6)	1609 (60.6)	<0.001	9552 (84.8)	1215 57.9)	<0.001
	Yes	1539 (14.4)	1048 (39.4)		1714 (15.21)	884 (42.1)	
**Planning suicide**	No	9414 (88.2)	1794 (67.6)	<0.001	9865 (87.6)	1363 65.3)	<0.001
	Yes	1257 (11.8)	858 (32.4)		1402 (12.4)	725 (34.7)	
**Attempting suicide**	No	7718 (93.2)	1694 (78.7)	<0.001	8115 (92.7)	1296 (76.6)	<0.001
	Yes	563 (6.8)	459 (21.3)		643 (7.3)	396 (23.4)	

**Table 4 children-10-00755-t004:** Adjusted logistic regression of the association of being bullied with the depressive symptom of sadness and hopelessness.

		Feeling Sad or Hopeless. AOR ^a^ (95% CI ^b^)
Bully types (at school and/or electronically)	Neither	Reference
	Either	2.43 (2.05–2.87) ***
	Both	3.46 (2.83–4.22) ***
Age	≤14 years old	Reference
	15 years old	0.80 (0.64–1.00)
	16 years old	0.80 (0.64–1.00)
	17 years old	0.71 (0.57–0.88)
	≥18 years old	0.77 (0.61–0.99)
Gender	Female	Reference
	Male	0.51 (0.45–0.58) ***
Race/Ethnicity	AIAN ^c^	Reference
	Asian	0.63 (0.25–1.59)
	Black	0.47 (0.19–1.11)
	NH/PI ^d^	0.21 (0.04–1.01)
	White	0.59 (0.25–1.39)
	Hispanic	0.57 (0.24–1.36)
	Multiple Hispanic	0.71 (0.30–1.68)
	Multiple non-Hispanic	0.97 (0.40–2.36)
Experienced sexual abuse	No	Reference
	Yes	2.83 (2.22–3.60) ***
Experienced physical dating violence	No	Reference
	Yes	3.12 (2.40–4.05) ***
Currently use alcohol	No	Reference
	Yes	1.18 (1.03–1.35) *
Currently used marijuana	No	Reference
	Yes	1.52 (1.31–1.75) ***
Sex first time before the age of 13 years	No	Reference
	Yes	1.14 (0.79–1.66)
Got ≥8 h of sleep per day	No	Reference
	Yes	0.46 (0.39–0.55) ***
Physically active at least 60 min per day on ≥5 days	No	Reference
	Yes	0.72 (0.64–0.80) ***
Used video/computer games or used computer ≥3 h per day	No	Reference
	Yes	1.68 (1.49–1.89) ***
Obesity	No	Reference
	Yes	1.24 (1.09–1.41) **

^a^ AOR: Adjusted odds ratio; ^b^ CI: Confidence intervals; ^c^ AIAN = American Indian or Alaska Native; ^d^ NH/PI = Native Hawaii or other Pacific Islander; * *p*-value < 0.05; ** *p*-value < 0.01; *** *p*-value < 0.001.

**Table 5 children-10-00755-t005:** Adjusted multinomial logistic regression of the number of bully types with suicidal tendencies.

No Suicidal Tendency (Reference)		Considering Suicide RRR ^a^ (95% CI ^b^)	Planning Suicide RRR ^a^ (95% CI ^b^)	Attempting Suicide RRR ^a^ (95% CI ^b^)
Bully types (at school and/or electronically)	Neither	Reference	Reference	Reference
	Either	1.59 (1.16–2.18) **	2.09 (1.64–2.65) ***	2.10 (1.60–2.74) ***
	Both	2.70 (1.96–3.73) ***	3.09 (2.40–3.98) ***	3.82 (2.92–4.99) ***
Age	≤14 years	Reference	Reference	Reference
	15 years	0.68 (0.43–1.08)	0.73 (0.52–1.03)	0.79 (0.55–1.14)
	16 years	0.88 (0.57–1.36)	0.71 (0.51–0.99) *	0.49 (0.33–0.71) ***
	17 years	0.86 (0.55–1.32)	0.63 (0.45–0.88) *	0.59 (0.41–0.85) ***
	≥18 years	0.77 (0.47–1.25)	0.76 (0.53–1.09)	0.53 (0.25–0.89) ***
Gender	Female	Reference	Reference	Reference
	Male	0.47 (0.40–0.55) ***	0.61 (0.50–0.75) ***	0.64 (0.51–0.81) ***
Race	AIAN ^c^	Reference	Reference	Reference
	Asian	0.17 (0.04–1.21)	1.52 (0.36–6.45)	1.52 (0.32–7.29)
	Black	0.23 (0.07–0.99)	0.70 (0.17–2.85)	0.94 (0.21–4.25)
	NH/PI ^d^	0.28 (0.02–2.01)	0.80 (0.08–8.23)	2.05 (0.24–17.2)
	White	0.32 (0.10–0.92)	1.09 (0.28–4.26)	1.10 (0.26–4.79)
	Hispanic	0.25 (0.08–1.03)	0.61 (0.15–2.47)	0.65 (0.14–2.97)
	Multiple Hispanic	0.26 (0.08–1.06)	0.83 (0.21–3.23)	0.89 (0.20–3.91)
	Multiple non-Hispanic	0.28 (0.08–1.37)	1.45 (0.35–5.94)	2.40 (0.53–10.76)
Experienced sexual abuse	No	Reference	Reference	Reference
	Yes	2.04 (1.41–2.94) ***	2.50 (1.87–3.35) ***	4.36 (3.29–5.78) ***
Experienced physical dating violence	No	Reference	Reference	Reference
	Yes	1.57 (1.04–2.37) ***	2.01 (1.47–2.74) ***	2.26 (1.65–3.10) ***
Currently use alcohol	No	Reference	Reference	Reference
	Yes	1.12 (0.86–1.45)	1.09 (0.88–1.34)	1.28 (1.02–1.62) *
Currently use marijuana	No	Reference	Reference	Reference
	Yes	1.77 (1.36–2.32) ***	1.78 (1.44–2.21) ***	2.29 (1.81–2.88) ***
Sex first time before the age of 13 years	No	Reference	Reference	Reference
	Yes	1.11 (0.58–2.09)	0.78 (0.45–1.36)	1.48 (0.94–2.34)
Got ≥8 h of sleep per day	No	Reference	Reference	Reference
	Yes	0.39 (0.26–0.58) ***	0.43 (0.32–0.58) ***	0.40 (0.28–0.56) ***
Physically active at least 60 min per day on ≥5 days	No	Reference	Reference	Reference
	Yes	0.55 (0.42–0.71) ***	1.03 (0.85–1.24)	0.63 (0.51–0.79) ***
Used video/computer games or used computer ≥3 h per day	No	Reference	Reference	Reference
	Yes	1.72 (1.36–2.18) ***	1.69 (1.40–2.03) ***	1.50 (1.22–1.84) ***
Obesity	No	Reference	Reference	Reference
	Yes	1.80 (1.42–2.29) ***	1.48 (1.21–1.79) ***	1.91 (1.55–2.35) ***

^a^ RRR: Relative risk ratios; ^b^ CI: Confidence intervals; ^c^ AIAN = American Indian or Alaska Native; ^d^ NH/PI = Native Hawaii or other Pacific Islander; * *p*-value < 0.05; ** *p*-value < 0.01; *** *p*-value < 0.001.

## Data Availability

A publicly available dataset was analyzed in this study and can be found at https://www.cdc.gov/healthyyouth/data/yrbs/data.htm (accessed on 29 April 2021).

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
