# Peer review of "Association of In-School and Electronic Bullying with Suicidality and Feelings of Hopelessness among Adolescents in the United States"

_children, 2023, doi:10.3390/children10040755_

Round 1

Reviewer 1 Report (Previous Reviewer 3)

1. This is an excessively explored database and topic, could not find much new on this.

2. There are serious issues with type 1 and type 2 errors.

3. Many crucial variables have not been investigated on suicidality.

4

Author Response

Reviewer # 1:

Extensive editing of English language and style required.

Response: We have had the manuscript reviewed thoroughly by a native speaker and incorporated the changes in the current version of the manuscript.

  1. This is an excessively explored database and topic, could not find much new on this.

We appreciate the critical observation. We have thoroughly searched the recent literature, and our statement about what this research still stands on its own: “Much emphasis has been given to bullying and mental health across literature, while less attention focuses on the association between bullying and suicide-related behavior.”

  1. There are serious issues with type 1 and type 2 errors.

The authors respect the reviewer’s observation. However, the statistical methods used in the study and the study protocol followed to assure that no false-positive occurs due to the authors’ false rejection of any true null hypothesis (Type I error) or falsely fail to reject any false null hypotheses based on our results using the determined threshold of p-values (a type II error). We used a p<=0.05 as a threshold for accepting hypotheses. We regret that we could not fully understand the nature of the issues spotted by the reviewer in this comment but will be happy to address it more if we can get more explanation.

  1. Many crucial variables have not been investigated on suicidality.

The authors agree with the reviewer that suicidality is highly complex and multifaceted, with many “crucial” contributing and facilitating variables. However, we set the scope for our study and investigated the variables within our scope. In this regard, we included the shortfall of limited factors as our study limitations.

Reviewer 2 Report (Previous Reviewer 1)

Dear Authors,

Thank you for the revised manuscript. I appreciate taking into account my comments and suggestions. One minor point I possibly missed during the first round of review: the MDPI does not support the APA style in numbers, thus the p-values should be corrected from < .001 to < 0.001 (Table 3).

Author Response

Reviewer # 2:

Dear Authors,

Thank you for the revised manuscript. I appreciate taking into account my comments and suggestions. One minor point I possibly missed during the first round of review: the MDPI does not support the APA style in numbers, thus the p-values should be corrected from < .001 to < 0.001 (Table 3).

Authors’ response:

We greatly appreciate the reviewer’s comments. As suggested, we corrected the p-values’ number format in Table 3.

Reviewer 3 Report (New Reviewer)

I read the paper by Tran Nguyen et al entitled "Association of in-school and electronic bullying with suicidality and feelings of hopelessness among adolescents in the United States". The topic is particularly interesting and topical the paper is well written but the results give us mediocre knowledge. Finishing the study of the manuscript, I wondered if the results of this study, with data from 2019, are valid today in 2023. Let me explain that I am not referring to the 4 intervening years but to the effect that the pandemic may have on the data. I would be delighted if the authors would give us a comparative study of the 2019-2020 data, even in a future paper. In this manuscript, I would suggest adding to the discussion or limitations of the study the potential impact of the pandemic, particularly on the epidemiological data of this study.

Another issue that the authors need to clarify is whether the authors were required to ask for a license to use that data.

Author Response

Reviewer # 3:

I read the paper by Tran Nguyen et al entitled "Association of in-school and electronic bullying with suicidality and feelings of hopelessness among adolescents in the United States". The topic is particularly interesting and topical the paper is well written but the results give us mediocre knowledge. Finishing the study of the manuscript, I wondered if the results of this study, with data from 2019, are valid today in 2023. Let me explain that I am not referring to the 4 intervening years but to the effect that the pandemic may have on the data. I would be delighted if the authors would give us a comparative study of the 2019-2020 data, even in a future paper. In this manuscript, I would suggest adding to the discussion or limitations of the study the potential impact of the pandemic, particularly on the epidemiological data of this study.

Another issue that the authors need to clarify is whether the authors were required to ask for a license to use that data.

Authors’ response:

We appreciate the reviewer’s concern about us analyzing the pre-pandemic data. The US Centers for Disease Control and Prevention (CDC) conducts the Youth Risk Behavior Surveillance System every other year. The most recent one, conducted in the fall of 2021, is not publicly available, making the 2019 YRBSS the closest pre-pandemic data available. The 2021 YRBSS preliminary results indicated significant increases in the percentage of youth who experienced persistent feelings of sadness or hopelessness, seriously considered suicide, made a suicide plan, and attempted suicide, while a decrease in the proportion of youth who were bullied at school and no change in the proportion of youth who were electronically bullied. The 2021 YRBSS was conducted in the fall of 2021, when most schools reopened regularly or intermittently after a long period of closure due to the pandemic. Perhaps, the decrease in the proportion of youth who were bullied at school is due to missing school, which may lead to increases in the percentage of youth who experienced persistent feelings of sadness or hopelessness, seriously considered suicide, made a suicide plan, and attempted suicide. Until we see a steady trend post-pandemic, the 2019 YRBSS may be a reliable dataset for the inferential statistic study to reflect normal realities and not realities amid the COVID crisis.

As suggested, we added to the discussion the potential impact of the COVID-19 pandemic.

The YRBSS data are available upon request from the CDC; thus, we are not required to have a license for use. www.cdc.gov/yrbss

Round 2

Reviewer 1 Report (Previous Reviewer 3)

thanks for the clarifications

This manuscript is a resubmission of an earlier submission. The following is a list of the peer review reports and author responses from that submission.

Round 1

Reviewer 1 Report

Dear Authors,

Thank you for the excellent paper. The topic is important and relevant to public health concerns. The paper is very well-organized. It was my pleasure to read it. Please consider my comments.

To better characterise study participants, please indicate a sample size, distribution (%) in gender groups and a mean (±SD) and range of age of the study participants in the abstract.

In the description of the sample characteristics (page 3, the first paragraph of the Results section), it would be helpful to provide the age range of the study participants.

I suggest providing a more detailed description of the covariates in the methods section with the response options. Some of them, like physical activity, time spent on computer games ≥3 hours a day and defining obesity, require references. Also, I wonder if the information on time spent on social networks and the extent of the activity of engagement in social networks were collected. If not, I also suggest adding this as a study limitation.

In tables 3 and 4, it would be more clear to replace -- into 1.00 or reference group (ref.).

The authors highlight the importance of social support from peers, teachers and parents as essential factors to prevent depression and suicidality in the introduction section. Nevertheless, this important aspect was not analysed in this study. Thus, I suggest adding this as an additional limitation.

Please list the main findings in the conclusion section to answer the study aim.

Reviewer 2 Report

Thank you for the opportunity to review this well written article. 

Some of the references are not current.

Reviewer 3 Report

1. This is a very poor quality paper with an overused database.

2. There is a large list of better papers from the same data-

https://scholar.google.com/scholar?hl=en&as_sdt=0%2C32&q=yrbs+bullying+hopelessness&btnG=&oq=yrbs+bullying+hop

3. The statistical models are not well discussed for fit, normality, etc.

4. There are no major implications for practice, research, prevention, policy, etc.

5. The social determinants of the outcomes are not discussed (e.g. parental violence, drug use, ACEs, etc)